# Rational Design by Structural Biology of Industrializable, Long-Acting Antihyperglycemic GLP-1 Receptor Agonists

**DOI:** 10.3390/ph15060740

**Published:** 2022-06-13

**Authors:** Lei Sun, Zhi-Ming Zheng, Chang-Sheng Shao, Zhi-Yong Zhang, Ming-Wei Li, Li Wang, Han Wang, Gen-Hai Zhao, Peng Wang

**Affiliations:** 1Key Laboratory of High Magnetic Field and Ion Beam Physical Biology, Hefei Institutes of Physical Science, Chinese Academy of Sciences (CAS), Hefei 230031, China; qiku500@mail.ustc.edu.cn (L.S.); shaochangsheng@hmfl.ac.cn (C.-S.S.); liwang@ipp.ac.cn (L.W.); wh_yingwang@163.com (H.W.); zhgh327@ipp.ac.cn (G.-H.Z.); 2Science Island Branch of Graduate School, University of Science and Technology of China, Hefei 230026, China; 3MOE Key Laboratory for Membraneless Organelles and Cellular Dynamics, National Science Center for Physical Sciences at Microscale, Division of Life Sciences and Medicine, University of Science and Technology of China, Hefei 230022, China; zzyzhang@ustc.edu.cn (Z.-Y.Z.); mwli@mail.ustc.edu.cn (M.-W.L.)

**Keywords:** glucagon-like peptide-1, GLP-1 receptor agonist, functional protein design, molecular dynamics simulation, long-acting antihyperglycemic

## Abstract

Glucagon-like peptide-1 (GLP-1) is easily degraded by dipeptidyl peptidase-4 (DPP-4) in the human body, limiting its therapeutic effect on type II diabetes. Therefore, improving GLP-1 receptor agonist (GLP-1RA) stability is a major obstacle for drug development. We analyzed human GLP-1, DPP-4, and GLP-1 receptor structures and designed three GLP-1RAs, which were introduced into fusion protein fragments and changed in the overall conformation. This modification effectively prevented GLP-1RAs from entering the DPP-4 active center without affecting GLP-1RAs’ ability to bind to GLP-1R, the new GLP-1RA hypoglycemic effect lasting for >24 h. Through molecular modeling, molecular dynamics calculation, and simulation, possible tertiary structure models of GLP-1RAs were obtained; molecular docking with DPP-4 and GLP-1R showed access to the fusion protein. The overall conformational change of GLP-1RAs prevented DPP-4 binding, without affecting GLP-1RAs’ affinity to GLP-1R. This study provides important drug design ideas for GLP-1RA development and a new example for application of structural biology-based protein design in drug development.

## 1. Introduction

Diabetes is a metabolic disease caused by the increased demand for insulin, and is characterized by the relative lack of insulin secretion, the absolute lack of insulin secretion, or insulin resistance, which causes the disorder of carbohydrates, fat, and protein metabolism and makes the patient’s blood glucose level higher than the standard value [1]. Incidence of diabetes has shown an increasing trend. The 9th edition of the Global Diabetes Atlas [2], which was released by the International Diabetes Federation (IDF) in 2019, shows that as of 2019, approximately 463 million people aged between 20 and 79 have diabetes (the vast majority are type II diabetes), accounting for 9.3% of the world’s total population in this age group. It is estimated that by 2030, the total number of diabetic patients will reach 578.4 million.

GLP-1 is derived from L cells of the intestinal mucosa and is a carboxy-terminated peptide chain sequence obtained by the cleavage of proglucagon by hormone proconvertase 1 (PC1) [3]. The primary function of GLP-1 is to promote the synthesis and secretion of insulin by pancreatic beta cells after binding to G-protein-coupled receptors on the cell membrane [4]. Some studies [5,6] have shown that GLP-1 expressed in pancreatic islet α cells can activate protein kinase A (PKA) and inactivate PKA-dependent N-type Ca^2+^ channels on the surface of α cells due to phosphorylation. Additionally, GLP-1 has been shown to reduce Ca^2+^ influx, inhibiting the exocytosis of α cells, thereby inhibiting the release of glucagon [7].

The half-life of natural GLP-1 is very short, and scientists have been working to find GLP-1RAs that resist dipeptidyl peptidase-4 (DPP-4) degradation. GLP-1 is the substrate of the proteolytic enzyme DPP-4, which cleaves and inactivates GLP-1 from the 8th amino acid (alanine) and is metabolized by the kidneys in urine [8]. The process from the production of GLP-1 to the final inactivation is extremely rapid, occurring within only 1.5–2 min [9,10]. To solve this problem, researchers hoped to obtain GLP-1 mutants that can resist DPP-4 enzyme degradation through amino acid point mutations. Interestingly, it was found that these mutations could severely weaken the binding ability of GLP-1 and its receptor, thus affecting the function of GLP-1 [11,12,13,14]. Mutations that can inhibit the cleavage of DPP-4 may more or less affect the binding of the mutant to GLP-1R. After various attempts to mutate wild-type GLP-1 to extend its half-life, more researchers have attempted to alter the molecular structure of GLP-1 or to bind the molecule to other proteins to reduce the effect of enzyme digestion. Although several GLP-1RA drugs have already been approved by the FDA, there are still various associated problems. Exenatide, which is derived from the Mexican Gila monster, is the first GLP-1RA approved by the FDA for marketing, but its similarity to human GLP-1 was found to be only 53%. It triggers an immune response and has many side effects [15,16,17]. Liraglutide and semaglutide have been chemically modified to connect the fatty chain to natural GLP-1 molecules and rely on the fatty chain to bind to albumin in the blood [18]. This reduces the rate of enzymatic hydrolysis and renal clearance of such drugs by DPP-4 and extends the biological half-life. However, the high price of chemical modification also hinders the widespread use of such drugs [19,20,21].

This study is based on the analysis of the tertiary structure of GLP-1, DPP-4, and GLP-1R, by way of adding anti-parallel β-sheet domains at the amino or carboxyl ends of GLP-1 to increase steric hindrance. Several GLP-1RAs were designed, and the relative spatial position between the antiparallel β-folded domain and GLP-1 molecule were adjusted by introducing intra-chain disulfide by cysteine mutation. In vitro interaction experiments have verified that these GLP-1RAs can improve the tolerance to DPP-4 while maintaining its affinity with GLP-1R. From the impaired glucose tolerance (IGT) mouse model, we found that one of the new GLP-1 analogs, G1, has a significantly longer action time in the animal body, and the control of the blood glucose level was greatly improved. Through molecular modeling and molecular dynamics calculations and simulations, we inferred the possible tertiary structure of G1. We simulated the binding of G1 to GLP-1R and DPP-4 using a molecular docking method, which preliminarily explained why G1 protein can maintain its affinity with GLP-1R and tolerate DPP-4. The above work provides important drug design ideas for the development of GLP-1RAs and provides a new example for the application of structural biology-based protein design in drug development.

## 2. Results

### 2.1. GLP-1 Receptor Agonists Design

To improve the duration of the hypoglycemic effect of GLP-1, we first investigated the spatial structure and binding modes of GLP-1, DPP-4, and GLP-1R. In this study, the N-terminal antiparallel β lamellar domain of superoxide dismutase (SOD) was selected as the fusion protein for GLP-1. The cross-sectional length of this lamellar structure is approximately 36 Å, which is larger than the 26 Å width of the substrate-binding pocket of DDP4. Meanwhile, the antiparallel β lamellar domain is structurally rigid and can provide an ideal steric hindrance. In addition, when GLP-1 analogs were bound to the DDP4 substrate pocket, the positive charge enrichment region in the antiparallel β lamellar domain could interact strongly with the negative charge enrichment region near the substrate pocket, which could further prevent GLP-1 analogs from entering the DDP4 reaction pocket. The active center and binding site of DPP-4 are both located in the narrow inner pocket of DPP-4. GLP-1 interacts with DDP-4 by extending the enzyme cutting site located at the N-terminal into the inner substrate pocket of DPP-4. By adding a rigid domain to the terminal of natural GLP-1 and forming additional steric hindrance near the enzyme active pocket of DDP4, we hoped to improve the tolerance of GLP-1RAs to DPP-4 digestion and extend the GLP-1RAs’ half-life.

According to the above design principles, we designed three GLP-1RAs with the sequence information shown in Figure 1 and as listed here.

G1: Compared with human GLP-1, Val33 is mutated into Cys33, followed by Gly36 with an 8-amino acid Linker sequence “GCGGGGGG” and the antiparallel β lamellar domain. Disulfide bonds are formed between Cys33 and Cys39.

G2: Compared with G1, the antiparallel β lamellar domain binds to the amino terminus of GLP-1 instead of the carboxyl terminus, and the disulfide bond can also be formed between Cys33 and Cys39, corresponding to G1.

G4: Compared with G1, Ser18 is mutated to Cys18, so the Cys18-Cys39 disulfide bond can be formed across a larger region, but without the link after Gly36.

The position difference of the disulfide bond will cause different degrees of bending of the peptide chain, resulting in different degrees of conformational changes.

To make the fusion protein, which is located at the carboxyl terminal closer to the substrate-binding pocket of the DPP-4 enzyme, we introduced a flexible linker between the natural GLP-1 sequence and β lamellar, which caused a certain waviness between the two domains. At the same time, cysteine was also inserted intramolecularly using the point mutation method. This was done in the hopes of promoting the bending of GLP-1 analogs through the formation of intramolecular disulfide bonds, to ensure that the fusion protein would be closer to the amino end of GLP-1 analogs in the spatial position. To avoid damaging the binding ability between GLP-1 analogues and GLP-1 receptor during the GLP-1 analog sequence design, we did not mutate any amino acids from positions 22 to 32, which are closely related to GLP-1 receptor binding.

### 2.2. GLP-1RAs Preparation

After the amino acid sequences of GLP-1RAs were determined, more suitable DNA sequences for the prokaryotic expression system of Escherichia coli were obtained using GeneOptimizer software. The codon adaptation index was 0.72 before optimization, and the adaptation index was increased to 0.99 after optimization.

The amplified plasmids in *E. coli* DH5α were sequenced, and the sequencing results that were consistent with the theoretical sequence and the extracted plasmids were then transferred into *E. coli* BL21 for expression. GLP-1RAs were cultured in liquid LB medium at 37 °C and expressed at 16 °C. All GLP-1RAs precursors existed in the form of inclusion bodies.

The inclusion bodies were renatured using a pH gradient and digested with trypsin at different concentrations to obtain target proteins. The G1 protein was purified using a G75 molecular sieve and C8-HPLC. The G2 protein was purified by Q-FF anion exchange, G75 molecular sieves, and C8-HPLC. The G4 protein was purified by CM-FF cation exchange, G75 molecular sieves, and C8-HPLC. The LC-MS results of the purified proteins are shown in Figure 2.

We studied the process of GLP-1RAs at a fermentation scale of 100 L, which indicated that these proteins had good industrialization potential. The expression system of *E. coli* is mature in industrial applications; it is stable and has a high fermentation yield. The dry cell weights of G1, G2, and G4 fermentation were 93, 89, and 95 g/L, while the dry weights of inclusion bodies were 3.6, 3.4, and 4.1 g/L, respectively. To reduce environmental pollution during the renaturation process, traditional urea, guanidine hydrochloride, and other strong denaturants were not used. The renaturation was operated at room temperature, and the energy consumption was also greatly reduced. G1 was purified by two-step chromatography with an overall yield of 69.6% in the purification process, and 382 mg of the target protein was obtained per liter of fermentation. The purification yields of G2 and G4 were 55.1% and 59.3%, respectively, and 286 and 315 mg of the target protein could be obtained per liter of fermentation, respectively.

### 2.3. GLP-1 Receptor Affinity

In the preliminary experiment, different concentrations of human GLP-1 were incubated with GLP-1R on the surface of INS-1 rat insulin cells embedded in 96-well plates, and then detected by the chromogenic reaction of the ELISA kit. When the concentration of human GLP-1 protein exceeded 1 µg/mL, the excessive human GLP-1 no longer interacted with the GLP-1 receptor on the surface of the embedded INS-1 rat insulin cells, and the result of the ELISA reaction reached its peak. The molar concentration of the GLP-1 receptor was approximately 303 nM at saturation, and the maximum absorbance at 450 nm was 3.550. The double reciprocal graph showed a linear relationship(log/log) between the concentration of human GLP-1 protein and the A450 measured by ELISA from 1 to 100 ng/mL.

Based on the results of the preliminary experiment, the concentrations of GLP-1RAs were diluted to 333, 100, 33, 10, 3.3, 1.0, 0.33 and 0.10 nM for ELISA. After the reaction, the dissociation constant (Kd) between the target protein and GLP-1 receptor was calculated by plotting [22,23,24] (Kd is equal to the ligand concentration when half of the receptors are bound by ligands; the smaller the Kd value is, the slower the dissociation is and the stronger the affinity is). The determination coefficient R^2^ was used to determine the fitting degree of the curve to the sample data. The higher the determination coefficient, the stronger the ability of the model to explain the dependent variable. The abscissa of the curve is the molar concentration of the measured protein, and the ordinate is $\frac{A_{0}-A_{i}}{A_{0}-A_{\infty }}\cdot {\left[x\right]}_{tot}$ [25], where *A*_0_ is the absorbance of the blank control at 450 nm, with an average value of 0.862; *A_i_* is the average value of the measured absorbance at 450 nm at each detection concentration; *A_∞_* is the absorbance at 450 nm at saturation, which is 3.550; and [*x*]*_tot_* is the approximately total concentration of the receptor (303 nM).

As shown in Figure 3, the R^2^ obtained by each group was above 0.95, and the curve fitted the sample data well. The Kd value of G1 was 4.953, which was significantly lower than that of human GLP-1 (11.12). The Kd value of G4 was 15.12, indicating that the affinity of G4 is similar to that of human GLP-1. The Kd value of G2 was 34.43, which was significantly higher than that of human GLP-1. The data indicated that the affinity of G1 and G4 for GLP-1R is not affected by structural adjustment and G1 is even better than human GLP-1. In conclusion, although G2 is similar to G1 and G4 in the introduction of the β-folding domain, direct fusion to the N-terminus of GLP-1 significantly affects its affinity to the receptor. However, when β-folding is added to the C-terminal, the conformational change induced by disulfide bonds does not affect binding with the receptor.

### 2.4. Stability Studies of GLP-1RAs In Vitro

To verify and compare the stability of GLP-1RAs in vitro, we attempted to digest GLP-1RAs in a PBS reaction system with an excess of DPP-4 enzyme, using human GLP-1 as the control substance. ELISA results, which were measured before adding the enzyme in each group, were used as the benchmark, and ELISA results obtained at different time points were compared with the benchmark to obtain the percentage of the residual. The residual percentages of GLP-1RAs at different time points were connected as curves, and the area under curve (AUCs) were calculated.

As shown in Figure 4, the residue of G2 under the action of DPP-4 enzyme is very close to that of human GLP-1, indicating that G2 has no inhibitory digestion ability. While G4′s curve is more stable, its AUC is 2.469, which is twice that of GLP-1 (1.025). G1 inhibited DPP-4 digestion much more significantly, with an AUC of 5.137, five times that of GLP-1, and approximately 20% of the protein remained after 24 h of digestion. This suggests that G1 has a much longer hypoglycemic effect.

### 2.5. Glucose Tolerance Test of GLP-1RAs on IGT Mice

After in vitro experiments were conducted to investigate the DPP-4 inhibition ability and receptor affinity of GLP-1RAs, further in vivo experiments were conducted to compare the hypoglycemic effects of GLP-1RAs on impaired glucose tolerance (IGT) mice. The model of IGT mice was established by a one-week injection of streptozotocin, compared with the normal glucose tolerance (NGT) group, the increase in blood glucose in the IGT group after intragastric glucose administration was statistically significant, and the increase in blood glucose in each experimental group was similar after grouping (as shown in Table A2). Each group was fasted for 9 h before the last administration, fasting blood glucose (FBG) was measured one hour after administration, and then an oral glucose tolerance test (OGTT) was performed. The GLP-1 group has a hypoglycemic effect, which is statistically different from the vehicle group (Figure 5a,b). Its AUC was 47.54, which was lower than that of the vehicle group (60.54) and slightly higher than that of the NGT group (40.57). The G2 group did not show a clear blood glucose control effect, and it was not statistically different from that of the vehicle group. The blood glucose curve of the G4 group was similar to that of the GLP-1 group; however, the hypoglycemic effect was better than that of the GLP-1 group at 60 min and 90 min, showing a better sustained hypoglycemic effect. The AUC of G4 was 43.39, which was between that of the NGT and GLP-1 groups. The G1 group had the most obvious hypoglycemic effect, and its blood glucose level was not only significantly lower than that of the GLP-1 group, but also significantly lower than that of the NGT group, with statistical differences within 120 min, and the AUC was 32.06, lower than that of the NGT group (40.57).

After fasting for 9 h, the OGTT was performed again the next day. The duration of GLP-1RAs was measured using the second day’s OGTT. As shown in Figure 5c,d, the blood glucose curves of the vehicle and GLP-1 groups were basically the same after the gavage of glucose on the second day, and there was no statistical difference except for at 120 min. The blood glucose level of the G2 and G4 groups at 30 min was slightly lower than that of the vehicle group, which was statistically different, and was comparable to that of the vehicle group at other time points. Combined with the area of blood glucose under the peak, the AUC of the G2 group was 60.78, which was not significantly different from the 62.99 of the vehicle group, while the AUC of the G4 group was 56.47, which was 12% lower than that of the vehicle group, indicating that G4 still had a certain hypoglycemic effect on the second day. The blood glucose level of the G1 group was lower than that of the vehicle group at each time point, and the difference was statistically significant. The blood glucose curve of the G1 group was close to that of the NGT group, and the AUC of the G1 group was 49.91, which is similar to the 45.74 of the NGT group, and significantly lower than that of the vehicle group, indicating that G1 can still achieve a significant hypoglycemic effect on the second day after administration. The blood sugar of IGT mice can be well controlled at a level equivalent to that of NGT mice, and the effect time of the drug was more than 24 h.

### 2.6. Molecular Modeling and Molecular Dynamics Simulation of G1

To elucidate the reasons for the excellent performance of G1 in DPP-4 degradation tolerance and glycemic control, we attempted to obtain its three-dimensional structural information by molecular modeling. The I-Tasser was first used to predict the tertiary structure of G1, and the scoring of the five models with the highest probability was as follows: Model1: C-score = −2.20, Model2: C-score = −2.60, Model3: C-score = −3.66, Model4: C-score = −2.72, and Model5: C-score = −5.00. The Tencent AI Lab was used to predict the 3D structure of the G1 protein, and it was found that the optimal 3D structure predicted was consistent with the Model3 structure predicted by I-Tasser. The above five models were evaluated using VERIFY 3D and ERRAT software, and model3 had the highest score. VERIFY 3D showed that 80.25% of residues had a 3D-1D score ≥ 0.2, and ERRAT scored Model3 as 91.78. The secondary structure of the G1 protein was obtained by far-ultraviolet infrared spectroscopy. The BeTeSel algorithm was used to evaluate the secondary structure of the G1 circular dichroism results, and the α-helical ratio was 25.7% while the antiparallel β-folding ratio was 16.1%. In Model3, the proportion of the secondary structure was 28.4% for α-helix and 17.4% for antiparallel β-fold, which is consistent with the results of the circular dichroism analysis.

To obtain a stable conformation of the G1 structure, we performed a 200 ns all-atom molecular dynamics simulation on Model3 using Gromacs software and obtained the optimal conformation after the simulation (Figure 6). RMSD values of Model3 are essentially stabilized at around 7.8 Å (Figure A1a). The RMSF values are largely stable at around 3.2 Å, except the terminal domain and loop of G31 to A41 and Q64 to P68 (Figure A1b). The RMSD and RMSF clearly showed the conformational stability (Figure A1).

### 2.7. Molecular Docking between G1 and GLP-1R and DPP-4

After obtaining the optimal stable conformation of G1 using the molecular dynamics simulation, we hoped to reveal the similarities and differences between GLP-1 and G1 when interacting with DPP-4 and GLP-1R by molecular docking. Zdock predicted the possible complex structures of GLP-1 and G1 with DPP-4 and GLP-1R, respectively. The most likely binding pattern of human GLP-1 to DPP-4 is shown in Figure 7a (highest binding score: 56.765). In this complex, GLP-1 binds well to the substrate pocket of DPP-4, which is consistent with the experimental fact that GLP-1 can be degraded by DPP-4. Since the antiparallel β-sheet structure prevents G1 from entering the substrate pocket of DPP-4, the most likely binding site of G1 and DPP-4 has moved to the outer region of the protein away from the active site of DPP-4 (highest binding score: 1381.959). Therefore, G1 is strongly resistant to the enzymatic effect of DPP-4 (Figure 7b).

Based on the crystal structure analysis of the GLP-1 complex and its receptor (PDB ID: 3iol), Gln_23_, Ala_24_, Ala_25_, Lys_26_, Glu_27_, Phe_28_, Ile_29_, Trp_31_, and Leu_32_ of GLP-1, and Leu_32_, Trp_39_, Asp_67_, Arg_121_, Leu_123_, Glu_127_, and Glu_128_ of GLP-1R are involved in the interaction between them (Figure 7c). By referring to the residues involved in the interaction above, the possible complex structure formed by G1 and GLP-1R was simulated (Figure 7d), which showed that G1 could still bind to GLP-1R in a binding mode similar to GLP-1 (the highest binding score: 910.159). The anti-parallel β lamellae, of the fusion expression of G1, have no ability to interfere with the binding process and impede the biological function of G1 in blood glucose control.

## 3. Discussion

Type 2 diabetes mellitus (T2DM) is a common metabolic disorder caused by persistent relative or absolute insulin deficiency, resulting in hyperglycemia [26]. GLP-1 is a peptide secreted by the pancreas that plays an important role in the glucose metabolic pathway, promoting insulin release in a glucose-dependent manner [27]. In addition, it also has the effect of reducing weight and improving insulin resistance [28,29]. However, GLP-1 can be easily hydrolyzed by DDP-4 and the short half-life of GLP-1 in vivo limits its clinical application.

In this study, we modified GLP-1 by stabilizing its conformation through disulfide bonds and attaching the fusion protein at the end of the molecule. This recombinant GLP-1 can be obtained by fermentation without organic synthesis. Previous studies have shown that GLP-1 modification gradually develops from amino acid mutations and partial amino acid modifications [12,30] to the overall structural modification. For example, some studies have fused GLP-1 molecules with human serum albumin [31], or PEG-modified GLP-1, to increase its molecular volume [32], while others have increased the half-life of GLP-1RAs by dimerization and antibody fusion [33]. Most GLP-1 modifications involve the expression of the basic GLP-1 fragment by organic synthesis or recombinant expression, followed by the addition of non-natural amino acid fragments, fatty acid chains, or other components that cannot be expressed by recombinant expression through solid phase synthesis, and then fusion with each other. The entire process is relatively complex, and there are many bottlenecks in process amplification during commercialization. In this study, the steric hindrance of the modified GLP-1RAs into the pocket of the DPP-4 enzyme substrate was increased, and therefore the degradation ability of the DPP-4 enzyme to GLP-1RA was reduced. The above modification did not interfere with the α-helix region of GLP-1, which is responsible for binding to the GLP-1 receptor, and it did not affect the biological function of GLP-1RA in controlling blood glucose.

In the in vitro enzyme digestion experiment, G1 showed far higher resistance to enzymolysis than human GLP-1 under the action of excessive DPP-4. Under the same digestion conditions, only 10% of human GLP-1 remained after one hour of digestion, while approximately 60% of G1 remained after one hour of digestion. The resistance of G4 to DPP-4 enzymatic hydrolysis was also greatly improved, and the accumulated residual area of G4 during the 24 h enzymatic digestion was more than twice that of human GLP-1. However, G2 showed no significant difference from human GLP-1, which means that it failed to meet the design expectations. Although the digestion site of DDP-4 in GLP-1 is located near the N-terminal, directly adding the fusion protein onto the N-terminal can theoretically maximize the steric hindrance near the active pocket of DDP-4, but this is not the case in practice. By designing a flexible linker, the β-folded sheet domain is introduced into the spatial region near the N-terminal, which can play a better role in strengthening the affinity.

In the affinity test, we found that the affinity of G4 to GLP-1R was similar to that of human GLP-1, indicating that the conformation of the G4 protein did not affect the binding of GLP-1R. However, the affinity of G2 for GLP-1R was much lower than that of human GLP-1. It is speculated that the β-folding at the N-terminal interferes with the binding of G2 to GLP-1R. The modified G1 has a 2-fold increase in affinity compared with wild-type GLP-1, which is rare in studies of GLP-1RAs. In the receptor affinity study of mono-PEGylated dimeric GLP-1 conjugates, it was found that when the molecular weight of the linked PEG increases, the affinity with the receptor continues to decrease. Furthermore, the same receptor affinity as wild-type GLP-1 can be maintained only when the molecular weight of PEG is less than 5 kDa; when the molecular weight of the PEG chain is expanded to 10 kDa or higher, the affinity between GLP-1RAs and the receptor will be significantly affected [32]. In this study, the molecular weight of the β-sheet was 4.3 kDa, and the presence of a flexible linker increased the spatial freedom of the β-folding sheet, which weakened the effect of the β-folding sheet on the binding site between GLP-1RAs and the receptor. In G2, the β-folded sheet is directly connected to the N-terminal, without a flexible linker as the transition, and the overall structure is relatively fixed in space. Therefore, the binding between G2 and the receptor is significantly affected. In addition, compared with long-chain PEG, the β-folded sheet itself has a certain spatial structure, which can form a positive charge enrichment region according to the surface potential analysis, which may play a role in the binding of G1 and G4 to the receptor.

Using in vivo experiments with IGT mice, we found that compared with human GLP-1, the G1 protein has a stronger glycemic control effect, and this effect can make the glucose of IGT mice reach the same level as normal mice, with a duration of more than 24 h. The G2 protein did not show a significant hypoglycemic effect in the IGT mice. The glycemic control effect of the G4 protein was almost the same as that of human GLP-1, but it maintained its hypoglycemic effect 24 h after medication. The results of the animal experiments were consistent with the results of the in vitro experiments on resistance to DDP-4 enzymatic hydrolysis and receptor affinity. G1, with higher receptor affinity and stronger resistance to enzyme digestion, showed a superior hypoglycemic effect.

In this study, although the in vitro test focused on the inhibitory effect of G1 on the enzymatic hydrolysis of DPP-4 was not significant, the IGT mice treated with G1 have the same blood glucose level or even better than NGT mice. Furthermore, the AUC of the blood curve was reduced by 50% compared with the vehicle and the hypoglycemic duration was more than 24 h, which was at the same level as the results of recent similar studies. The modified G1 receptor affinity was significantly higher than that of the wild type. Among the recently reported modifications of GLP-1RAs, other approaches have been used to enhance the effect of controlling blood sugar. One study fused GLP-1 with anionic aromatic small molecules, which acts like an adipose side chain and can conjugate albumin. The AUC of the blood curve was reduced by approximately 60% compared with that of the vehicle [34]. In another report, individual residues in the α-helix of wild-type GLP-1 were replaced with three consecutive ureido residues, and the AUC of the blood curve was reduced by approximately 60% compared with that of the vehicle [35]. The results suggest that, in addition to enhancing the resistance to enzymatic hydrolysis and prolonging the half-life of GLP-1RAs, enhancing the affinity between GLP-1RAs and receptors is another possibility for enhancing their hypoglycemic ability.

Using molecular modeling and docking, we predicted the 3D structure of the G1 protein and the binding mode of G1 to DPP-4 and GLP-1R. To further understand the effects of modification on the G1 protein, the most likely three-dimensional structure model of the G1 protein was obtained by combining the three-dimensional structure prediction, the evaluation of secondary structural elements, and a 200 ns total atomic molecular dynamics simulation. Based on the interaction information of human GLP-1 with the DPP-4 enzyme and GLP-1R, we simulated the possible binding patterns of the G1 protein with DPP-4 and GLP-1R by molecular docking. The docking results show that the binding mode of G1 to GLP-1R was consistent with that of GLP-1 and GLP-1R. Meanwhile, G1 and DPP-4 could not obtain a reasonable binding model through molecular docking; the binding site was far away from the active site of DPP-4. These results reflect the structural basis of G1 protein’s ability to tolerate DPP-4 enzymatic hydrolysis and maintain its binding ability to the GLP-1 receptor.

Compared with other GLP-1RAs modified by PEG modification, glycosylation modification, and the addition of aliphatic chains, G1 is more convenient in preparation and can be obtained directly using *E. coli* fermentation, which avoids organic synthesis and subsequent modification, reduces by-products and impurities, and lowers the production cost. The structural differences of G1, G2, and G4, as well as the corresponding subsequent experimental results, suggest that the selection of disulfide bond positions will have different influences on the overall bending degree of GLP-1, thus affecting the affinity and function of GLP-1RAs and GLP-1R. At the same time, the selection of C-terminal fusion proteins is worth further exploration, as the differences in hydrophobicity and different structures of fusion proteins will also play a key role in the overall function of GLP-1Rs. Based on the above optimization and combination on the basis of G1, we expect to design drug molecules with longer half-lives and more hypoglycemic effects in the future.

## 4. Materials and Methods

### 4.1. GLP-1RAs Preparation

The designed amino acid sequence of each GLP-1RA was analyzed using GeneOptimizer [36] or gene expression analysis, and multi-parameter gene optimization. The synthesized DNA plasmids (Jixiang Biology, Hefei, China) were double-digested with NdeI and Bpu1102I, and pET-24a(+) plasmid vector fragments were treated with the same double digestion. Plasmids and vectors were ligated at 22 °C for 1 h under the catalysis of T4 ligase, then transformed into competent *E. coli* DH5α and amplified in LB medium containing kanamycin sulfate. After that, the plasmids were extracted from *E. coli* DH5α and transformed into *E. coli* BL21 for expression. After fermentation and low-temperature induction with 1 mM IPTG at 16 °C, the inclusion bodies were homogenized and refolded into the correct tertiary structure. After refolding, the GLP-1RA precursors were protected with citraconic anhydride at pH 9.0, and the excess citraconic anhydride was neutralized with 5 mM ethanolamine. Following this, the protected GLP-1RA precursors were digested with trypsin in different proportions to obtain target proteins. Citraconic anhydride bound to lysine dissociated at pH 2.5. GLP-1RAs were first separated by semi-preparative C8-HPLC, and the leader peptide and residual trypsin were removed under acidic conditions and then purified using a G75 molecular sieve, Q FF anion exchange chromatography, CM FF cation exchange chromatography, etc. During the digestion and purification process, LC-MS was used to confirm the molecular weight and locate the target protein.

### 4.2. Affinity Test

Commercialized human GLP-1 was used as a positive control (ZhongTai Biochemical, Hangzhou, China). The binding affinity of GLP-1RAs to the GLP-1 receptor on rat insulinoma cells (Fuxiang Biotechnology, Shanghai, China) was measured using a human GLP1 (7–36) ELISA kit (Abcam, Waltham, MA, USA). To determine the optimal response concentration, INS-1 rat insulinoma cells were embedded in a blank 96-well plate and human GLP-1 protein was used to test the binding ability of GLP-1R on the surface of INS-1 rat insulinoma cells through preliminary experiments. According to the reaction steps of the ELISA kit, using PBS as a blank control, each GLP-1RA of different concentration was combined with GLP-1R on the surface of the embedded INS-1 rat insulinoma cells, eluting unbonded GLP-1RAs. The combined GLP-1RAs were then subjected to a chromogenic reaction, and the corresponding absorbance was measured. Each concentration consisted of three sets of parallel samples. The measured data were fitted using GraphPad Prism software (version 8.0) (Harvey Motulsky, San Diego, USA) using a single-site affinity method to obtain the dissociation constant Kd and decision coefficient R^2^.

### 4.3. Enzymatic Hydrolysis

The degradation rate of each GLP-1RA reacted with DPP-4 under the same reaction conditions was measured using an ELISA kit, with human GLP-1 as a control. The excess DPP-4 and GLP-1RA were incubated at 30 °C in a pH 7.4 PBS buffer, at different reaction time points. Samples were taken to measure residual GLP-1 using an ELISA kit. The absorbance of the color reaction between the target protein and the GLP-1 antibody embedded in the 96-well plate were drawn using GraphPad Prism software (version 8.0) (Harvey Motulsky, San Diego, USA) with time as the X-axis and the residual percentage of GLP-1RA relative to the initial value as the Y-axis. The area under the curve of human GLP-1 and GLP-1RAs reacted with excess DPP-4 under the same conditions was taken as a representation of their hypoglycemic effect and duration in vivo.

### 4.4. Glucose Tolerance Test in Model Mice

Male KM mice age 6–8 weeks with weight of 18–25 g were selected and injected with STZ at a dose of 40 mg/kg per mouse to establish mouse model of IGT. One week after the injection, OGTT was performed. Blood was collected from the tail tip of each mouse to measure the blood glucose concentration. Mice with blood glucose level increase greater than 200% after intragastric administration of glucose were selected. The mice were divided into the following groups: vehicle, GLP-1, and GLP-1RAs (G1, G2, G4) groups, with 8 mice in each group. A group of 8 mice, which did not receive treatment, was added as the normal glucose tolerance (NGT) group. The NGT and vehicle groups were treated with normal saline, and the other groups were treated with the corresponding protein solution diluted to 20 µmol/L with normal saline. Each mouse was injected with 25 µL/10 g body weight. The FBG of each group was measured one hour after the last administration, and then 2.5 g/kg glucose solution was administered by gavage immediately, and blood glucose was measured at 30, 60, 90, and 120 min after gavage. The FBG was measured the same time the next day, and the glucose solution was administered by gavage again at 2.5 g/kg. Blood glucose was measured 30, 60, 90, and 120 min after gavage. The statistical software GraphPad Prism 8.0, was used to process the data. The blood glucose level at each time point was compared between the groups using two-way analysis of variance. *p* < 0.05 was considered statistically different.

### 4.5. Molecular Modeling

I-TASSER [37] and the tertiary structure prediction tools of Tencent AI lab “Available online: https://drug.ai.tencent.com/console/cn/tfold (accessed on 5 February 2021)” were used to predict the G1 structure from the G1 protein with the best results in the functional experiments. The top five structural models with the highest scores from the I-TASSER program were compared with the optimal structure generated by tfold. VERIFY 3D [38,39] and ERRAT [40] were used to evaluate the predicted models and eliminate unqualified models. The secondary structure of the qualified model was compared with the circular dichroism result, and a reasonable structural model whose secondary structure is in accordance with the circular dichroism result was obtained. The molecular dynamics simulation was performed using the G1 model to obtain a stable conformation that may exist in the solution.

### 4.6. Molecular Dynamics Simulation

The best result, Model3, was used as the initial structure, and the molecular dynamics simulation was performed using GROMACS-4.5.5. The protein force field used was the AMBER99SB [41] force field. The protein was placed in the center of a regular dodecahedron box, with a distance between the protein and the box boundary being no less than 1.2 nm. Following this, 12,413 TIP4Pew [42] water molecules, 35 sodium ions, and 35 chloride ions were added to the box, which ensured that the total charge of the system was 0 while the salt concentration of the system reached 140 mM. The steepest descent and conjugate gradient methods were used to optimize the energy of the system to ensure that the topological conformation of the system reached the optimum before the simulation. Before the simulation, a 100 ps constraint kinetics simulation was performed to constrain the protein with a constraint force of 1000 KJ mol^−1^ nm^−2^. The P-LINCS algorithm [43] constraint method was used for all chemical bonds. The two-way cutoff radius calculated by van der Waals force was 0.9 and 1.4 nm, the adjacency list was updated every 20 fs, and the PME algorithm was used to calculate of long-range electrostatic interaction [44], where the interpolation order was set to 4 and the tolerance was set to 10^−5^. The time length of the molecular dynamics simulation was 200 ns, the step length of the molecular dynamics simulation was 2 fs, and the integration algorithm used was the leapfrog algorithm [45]. To ensure the molecular dynamic simulation was carried out under NPT conditions, the temperature of the protein, water, and ions in the system were coupled to 300 K using the speed re-adjustment algorithm, the relaxation time was 0.1 ps, the pressure of the system was coupled to 1 bar, the relaxation time was 0.5 ps, and the compressibility was 4.5 × 10^−5^ bar^−1^.

### 4.7. Molecular Docking

The complexes of DPP-4 and human GLP-1, DPP-4 and G1, GLP-1R, and G1 were molecularly docked using zdock3.0.2 software [46]. According to the experimental data [47], for DPP-4, Glu_205_, Glu_206_, Ser_630_, Tyr_631_, Val_656_, Trp_659_, Tyr_662_, Tyr_666_, Asp_708_, Val_711_, and His_740_ were selected as the active sites, GLY_10_ was selected as the active site for GLP-1, and corresponding GLY_4_ was selected as the active site for G1. For the docking of GLP-1R and G1, according to the interaction analysis of the crystal structure of GLP-1R and GLP-1 [48], Leu_32_, Trp_39_, Asp_67_, Arg_121_, Leu_123_, Glu_127_, and Glu_128_ were selected as the active residues of the GLP-1 receptor, and Gln_17_, Ala_18_, Ala_19_, Lys_20_, Glu_21_, Phe_22_, Ile_23_, Trp_25_, and Leu_26_ were selected as possible active residues of G1. Other docking parameters used were the default zdock parameters.

## 5. Conclusions

In summary, based on the analysis of the three-dimensional structure of the protein, we introduced an antiparallel β-sheet into the natural GLP-1, which prevented the GLP-1RA from binding to the DPP-4 substrate pocket without compromising its affinity with GLP-1R. Thus, the tolerance of the analogues to DPP-4 enzymatic hydrolysis in vivo and in vitro was significantly improved. Through a glucose tolerance test in IGT mice, we found that the G1 protein has a much better glucose control effect and a much longer duration than natural GLP-1. Although the molecular dynamics simulation and molecular docking experiments have preliminarily explained the reasons why the G1 protein has the above advantages over natural GLP-1, the safety and pharmacokinetic characteristics of the G1 protein as a potential drug remain to be further studied and confirmed. Our work provides an important design idea for the development of GLP-1RAs and provides a new example for the application of structural biology in drug development.

## Figures and Tables

**Figure 1 pharmaceuticals-15-00740-f001:**
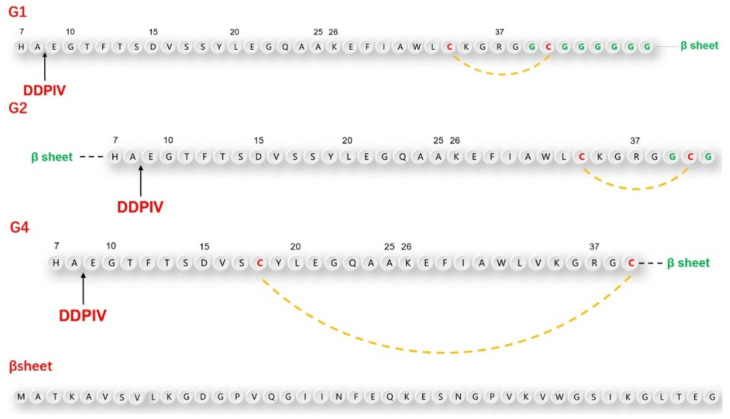
Sequences of GLP-1Rs (G1, G2, G4), β sheet is the antiparallel β lamellar domain added on N-terminal or C-terminal.

**Figure 2 pharmaceuticals-15-00740-f002:**
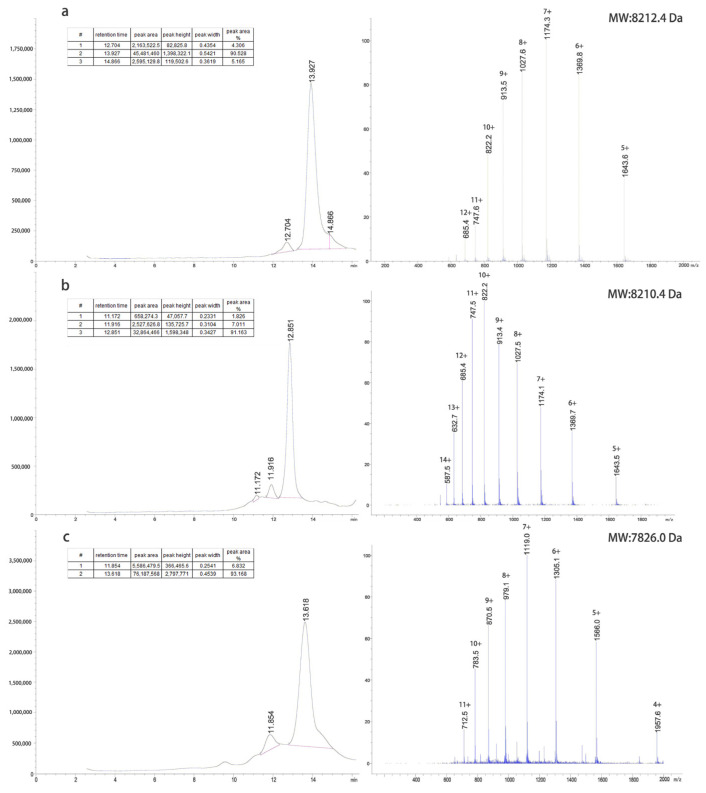
GLP-1RAs preparation. The refolding, digestion and purification processes were monitored by LC-MS, and the results of each GLP-1RAs obtained after purification were as follows: (**a**) G1 MW: 8212.4 Da; (**b**) G2 MW: 8210.4 Da; (**c**) G4 MW: 7826.0 Da.

**Figure 3 pharmaceuticals-15-00740-f003:**
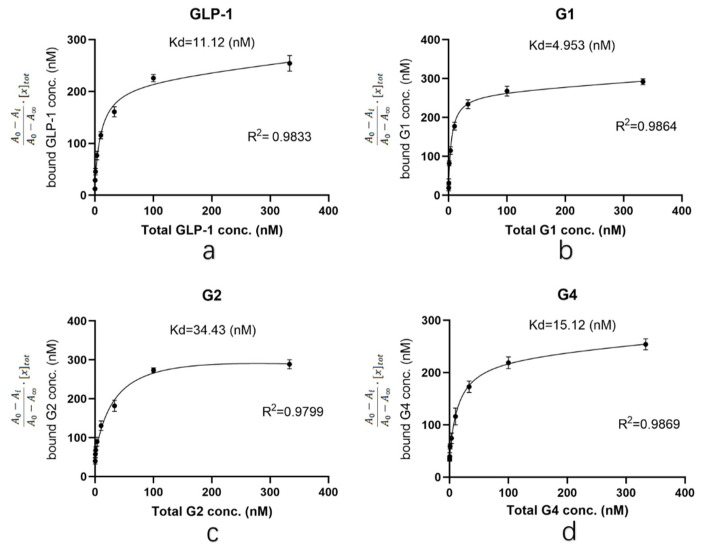
Affinity between GLP-1 receptor and (**a**) GLP-1; (**b**) G1; (**c**) G2; (**d**) G4.

**Figure 4 pharmaceuticals-15-00740-f004:**
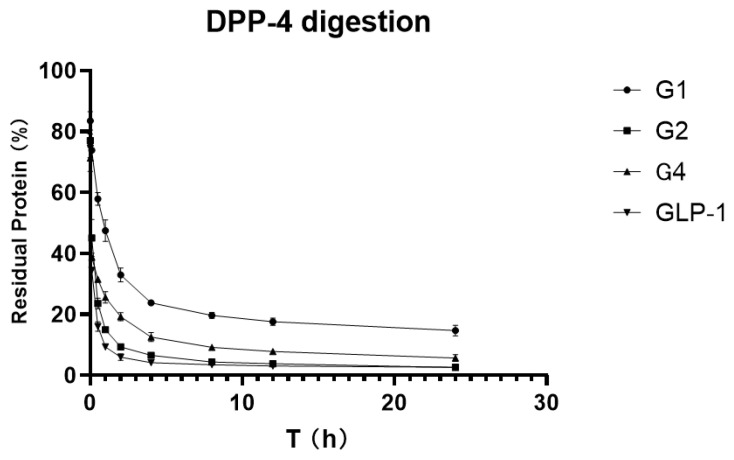
DPP-4 digestion curve. See also Table A1: Area Under Curve of DPP-4 digestion.

**Figure 5 pharmaceuticals-15-00740-f005:**
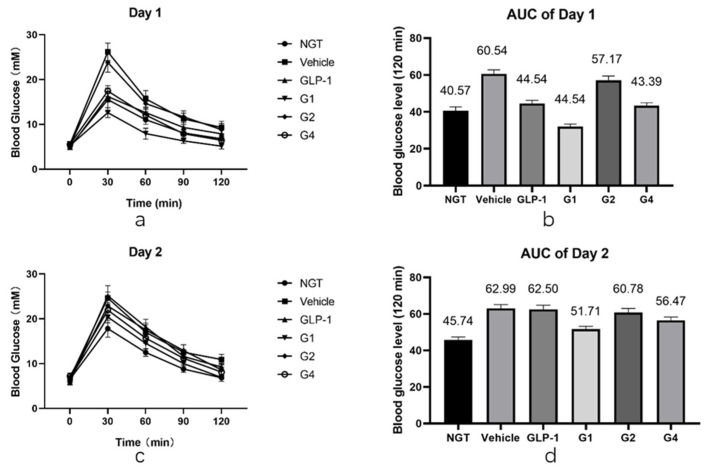
Glucose tolerance test of GLP-1RAs. (**a**) Blood glucose curve on the first day. (**b**) Blood glucose AUC on the first day. (**c**) Blood glucose curve on the second day. (**d**) Blood glucose AUC on the second day. See also Table A3: “The first day’s Glucose tolerance test results” and Table A4: “The second day’s Glucose tolerance test results”.

**Figure 6 pharmaceuticals-15-00740-f006:**
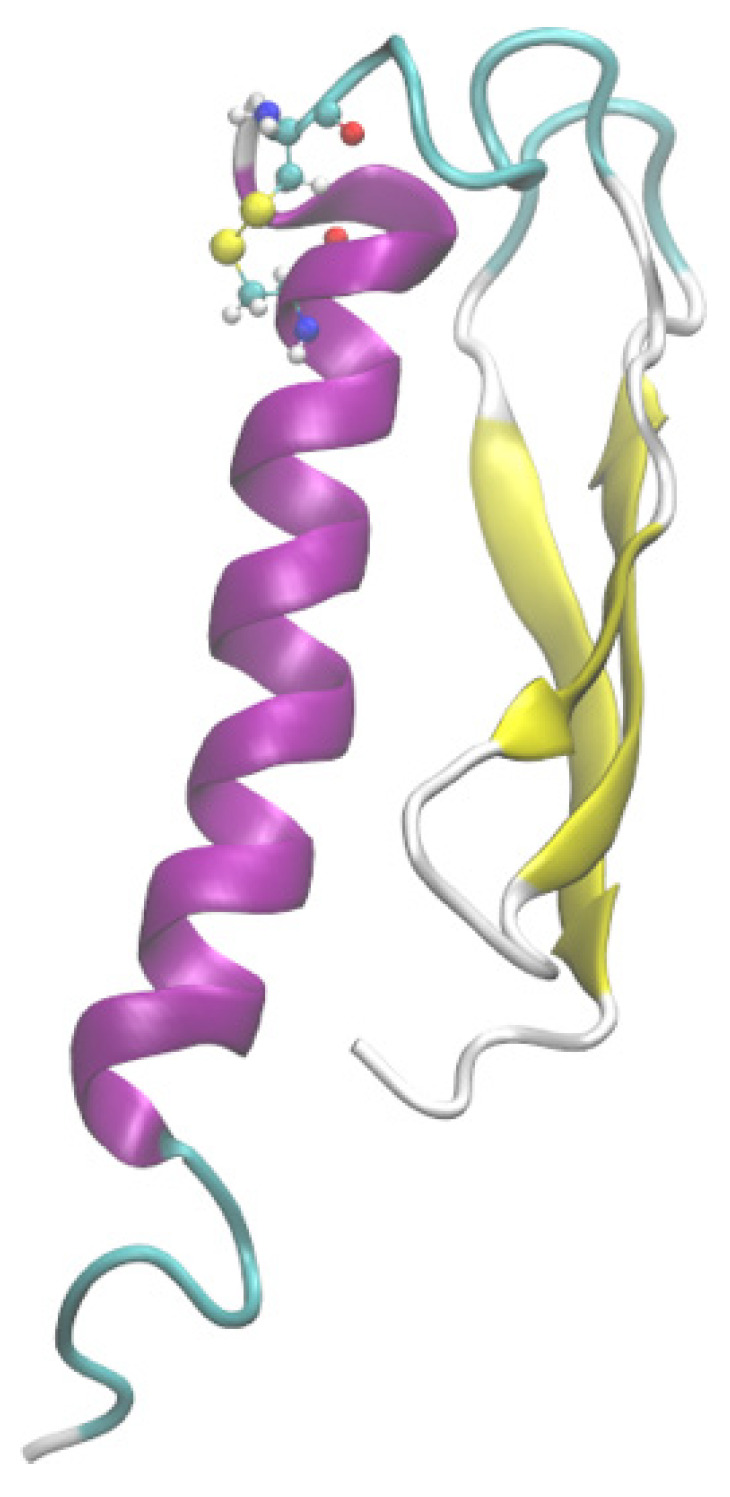
G1 modeling structure (Model3).

**Figure 7 pharmaceuticals-15-00740-f007:**
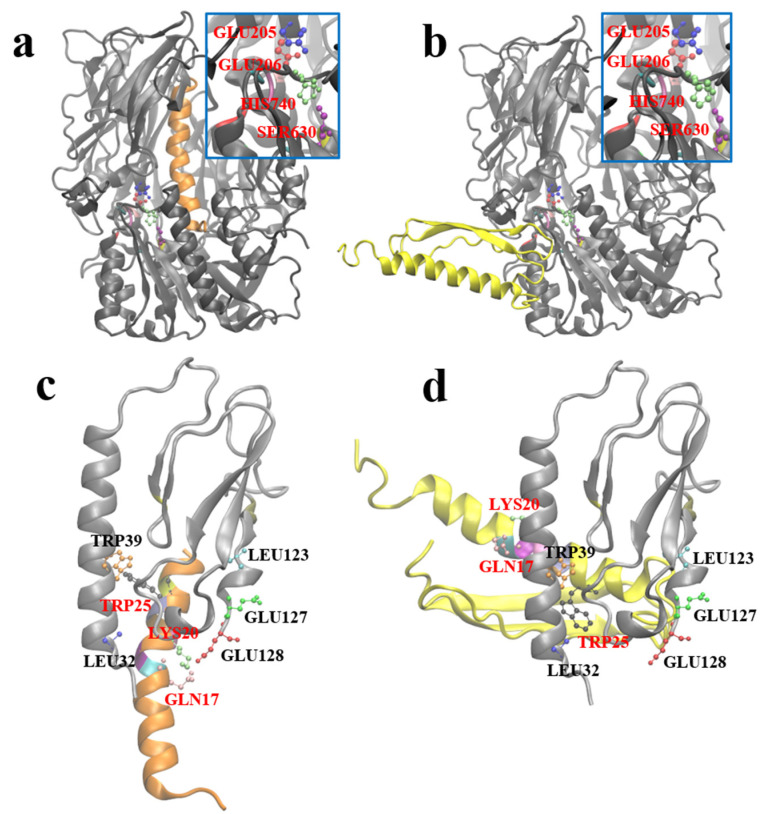
Molecular docking structures. (**a**) GLP-1 and DPP-4 molecule complex. (**b**) G1 and DPP-4 molecule complex. (**c**) GLP-1 and GLP-1R complex. (**d**) G1 and GLP-1R complex. The DPP-4, GLP-1, and G1 GLP-1R are colored by gray, orange, yellow and sliver, respectively. In DPP-4, the represented active residues GLU205, GLU206, SER630 and HIS740 are colored by blue, red, lime and purple, respectively. In GLP-1, the represented active residues GLN17, LYS20 and TRP25 are colored by pink, lime and black, respectively. In GLP-1R, the represented active residues LEU32, TRP39, LEU123, GLU127 and GLU128 are colored by blue, orange, cyan, green and red, respectively. The represented active residues are highlighted by CPK model, and other active residues are showed by New Cartoon model.

## Data Availability

The data presented in this study are available on request from the corresponding author. The data are not publicly available due to partner want to continue the research.

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
