# Peer review of "Rational Design by Structural Biology of Industrializable, Long-Acting Antihyperglycemic GLP-1 Receptor Agonists"

_pharmaceuticals, 2022, doi:10.3390/ph15060740_

Round 1

Reviewer 1 Report

In their article, Sun et al. proposes a study of a modified peptide of Glucagon-like peptide1 (GLP1), promoting the synthesis and secretion of insulin within the body. Because of the short life-time of GLP1, Authors have proposed modified peptides in order to increase this life-time. The study combined exmeprimental data and theoretical approaches.

In my opinion, this article deserves to be published in Pharmaceuticals, but first some improvements have to be made. I want to precise also that I will focus only on the theoretical experiments, since I am not an expert regarding the experimental part.

1) Regarding the MD simulations of G1, it misses in the paper RMSD and RMSF analysis. Indeed, one snapshot of a structure is insufficient to say that the conformation is at equilibrium and very stable, especially in the case of a 100 ns MD, which is correct (small system) but not state of the art regarding sampling statistics. Those analysis could be inserted in appendix instead of the main text in order to assess the comments of the authors.

2) I have some questions regarding the molecular docking: authors have decided to keep the DPP-4 and GLP-R1 recognizing residues of GLP-1 for the calculation with G1. Conservation of active residues have also be made for G1 (by transposition) and GLP-1. I think that will bias the calculation for G1, because in this way, authors consider that the addition of the anti-parallel beta sheet has no impact on the binding, nor a possible involvement (and maybe reinforcement ?) in the interaction.  I suggest that the authors must realize blind docking in order to corroborate their reults obtained with guided docking.

3) Regarding Figure 7a, I don't understand why the active residues of DPP-4 are quite far away of the binding pose of GLP1. Is this a mistake or not ? Generally, the figure 7 is quite difficult to read, especially with the position of the active residues. Please rework fig 7A and 7B.

Minor revision:

-There are few typos alla long the text and figures, please correct it. Few examples below

l284: the name Gromacs is with an "s"

Fig5 : There is an "Alt+A" box in fig 5C. Remove it please.

l501 : I think that the force field chosen is AMBER99SB and not '9SB'

-If possible, I suggest the author to continue by adding 100ns more for the MD simulation of G1.

Reviewer 2 Report

The overall presentation is reasonable. Improvement with added reference in the introduction, results, and discussion section is necessary. Figure 2 and figure 7 are two critical figures that must be improved from the current version.

Reviewer 3 Report

The manuscript describes the design and characterization of new GLP1-RAs. In general, the work will be of interest to those involved in antidiabetic drug design and specially to those working with GLP1 receptor agonists. However, there are some points to be considered before its acceptance for publication.

1.    In affinity experiments (Figure 3), there are some inconsistences:
a)    To determine Kd it is important to reach the saturation and this is not clear in the graphs, it seems like a higher concentrations are needed.
b)    No matter the R2 value is acceptable, some Kd values has no sense, for example in subsection “c” a Kd value of 14.44 nM is obtained, how is this possible if the highest concentration used was 13 nM?
c)    In the same context, in subsection “b” the Kd value was 0.58 nM, how the authors explain that at 13 nM the saturation was not reached?
d)    In general, this part of the results needs to be checked carefully, additionally, Kd values need units, in this case “nM”. Furthermore, would be useful to include in methodology the equation used to adjust the data.
2.    In molecular modeling (Figure 6), the image does not show the formation of intramolecular disulfide bond, what kind of evidence supports that the structural folding of G1, in solution, does not make this bond? if the idea originally was to perform the point mutations to encourage the formation of disulfide bond? Nothing is mentioned about this and needs to be discussed because it changes all the computational studies.
3.    Finally, a small suggestions, in line 39 the word “sugar” should be replaced by “carbohydrates” and in line 40 the word “sugar” should be replaced by “glucose”.

Round 2

Reviewer 1 Report

The paper has been improved since last version and it is now ready for publication in Pharmaceuticals.

I just have one remark for the authors: please modify the length of the simulation in the manuscript (it is still written100 ns, but Authors have improved it to 200)

Author Response

Reply: Thanks for the suggestion. The length of the simulation in the manuscript was revised.

Reviewer 2 Report

Figure 2 remains unreadable. Authors may consider identifying major bands and a summary table in the form of figure captions.

Figure 7 should be improved. Consider moving the legends.

The authors have added some new text to improve the quality of the manuscript. 

Reviewer 3 Report

The manuscript was corrected according to the suggested.

Author Response

Reply:Thanks a lot